

# Reflection seismic investigations on south Gotland, Sweden, to evaluate CO₂ storage strategies

Christopher Juhlin[1], Mikael Erlström[2], Peter Hedin[3], Bojan Brodic[4], Daniel Sopher[3]

[1]Uppsala University, Dept. of Earth Sciences, Villavägen 16, 75236, Uppsala, Sweden
[2]Lund University, Dept. of Geology, Sölvegatan 12, 223 62 Lund, Sweden
[3]Geological Survey of Sweden, Villavägen 18, 75236, Uppsala, Sweden
[4]Comprehensive Nuclear-Test-Ban Treaty Organization - CTBTO, Vienna, Austria

*Corresponding author: christopher.juhlin@geo.uu.se*

**Abstract.** Reflection seismic data were acquired in the Sudret area of Gotland between the 6th and 13th of November, 2023. Objectives of the survey were to obtain images of the subsurface down to the Precambrian basement in the vicinity of two cored boreholes that had been drilled earlier down to about 800 m. The seismic profiles were positioned to provide a better understanding of the sedimentary strata and local structure near the two boreholes. It was also hoped

that they could be used to correlate the properties of the geological formations offshore for studying the potential of future geological storage of CO₂ within Swedish waters. For these purposes a sparse 3D survey was acquired that covered a c. 300 m by 700 m rectangular area with high fold, including the locations where the boreholes were drilled. A longer c. 2.8 km 2D profile was also acquired adjacent to the 3D survey that ran roughly in a N-S direction. In addition, distributed acoustic sensing (DAS) measurements were performed in the two cored boreholes. We report here

on some results from the 2D and 3D surveys and from the DAS measurements, incorporating information from the core and sonic logs.

Numerous semi-continuous reflection horizons are observed in the c. upper 500 ms after stacking. A particularly strong reflection at about 330 ms likely originates from the top of Ordovician limestones. Generation of synthetic seismograms based on the acquired sonic logs in the two boreholes confirms this interpretation. Cambrian sandstones are also

reflective, as well as shallow sandstone layers in the upper 150 ms. Normal moveout (NMO) velocities are relatively constant at about 3500 m/s. However, depth conversion using this velocity places the reflectivity deeper than is expected from the well data. In comparison, using the DAS data, the vertically propagating P-wave velocity can be measured at an average 3100 m/s from surface to 580 m depth. Using this velocity for depth conversion provides more reasonable depths to the main horizons. Since the NMO velocities are largely controlled by the horizontal velocity of

the rock the difference between these and the DAS velocity can be explained by the rocks in the area having significant anisotropy. Seismic modeling indicates that a horizontal velocity of about 3500 m/s is necessary to explain the difference between the NMO velocity and the vertical velocity. This corresponds to an anisotropy of about 13%. This may be important to take into account when acquiring and processing future, or vintage, offshore seismic data for the purpose of mapping potential structures or formations for CO₂ storage.






## 1 Introduction

Geological storage of $CO_2$ has for several decades now been considered an option for reducing greenhouse gas emissions. In particular, the capture and storage of $CO_2$ from large point sources, such as coal fired plants and cement production facilities, has the potential to significantly reduce emissions. However, there are few active projects

worldwide where this is taking place. As of 2023 only about 40 Mt of the c. 36 Gt of $CO_2$ that is emitted annually, was captured (Dziejarski et al., 2023). Although the amount actually captured today is small there is an increasing interest in applying the technology with a significant number of projects in the planning stage (Martin-Roberts et al., 2021). Therefore, several European countries are re-evaluating their storage potential, including Sweden. Considering the aforesaid, in 2023 the Swedish government gave the Geological Survey of Sweden (SGU) the task to study the potential

for $CO_2$ storage within Swedish territory.

During initial studies on potential storage areas in Sweden two areas were identified as most prospective, the southern Baltic Sea and southwestern Sweden (Erlström et al., 2011). Currently, Swedish legislation only allows smaller amounts of $CO_2$ (<100 kt) to be stored geologically onshore, therefore any large scale storage would have to be offshore. However, drilling onshore is significantly cheaper than drilling offshore, so if suitable onshore sites can be identified as

analogs to the offshore sites then these sites can provide valuable information about the properties of the reservoir and cap rocks. As part of the SGU program it was decided to first core drill at one site on the southern part of Gotland island in the Swedish part of the Baltic sea and then the following year at another site onshore in southwestern Sweden. For both sites it was considered important to acquire reflection seismic data to be able to extrapolate drilling results away from the boreholes and investigate if nearby faults are present.

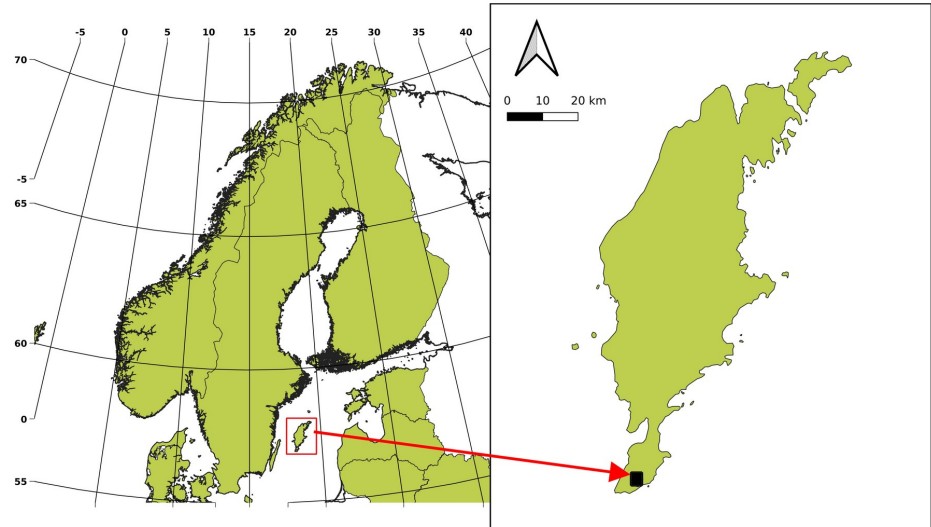

**Figure 1. Location map. The red arrow shows the location of the survey area on southern Gotland.**

In southernmost Gotland (Fig. 1), in particular the Sudret area, the target reservoir contains the Middle Cambrian Faludden and Lower Cambrian När and Viklau sandstone units (Fig. 2). These occur at depths of between c. 580 m and 800 m on Gotland, depths which are typically considered to be too shallow for injected $CO_2$ to remain in the supercritical state. No injection is currently planned at Sudret, but successful shallow $CO_2$ injection test sites exist, for example Ketzin, Germany (Förster et al., 2007). Hence, the Sudret area could be suitable for a small scale test site in the



future, as at Ketzin. Probably the greatest benefit and geoscientific contribution of the Sudret site is the possibility to improve the assessment of the reservoir and its suitability as a potential $CO_2$ storage alternative in the offshore areas southeast of Gotland. As the Cambrian Sandstone is considered the main candidate for storage in the Baltic region there is a great need for cost effective feasibility studies, such as land based investigations before going offshore. Aside from $CO_2$ storage, the reservoirs below Gotland are also of interest from the standpoint of geological energy storage and

geothermal energy (Sopher et al., 2019) where the results from this study can also contribute.

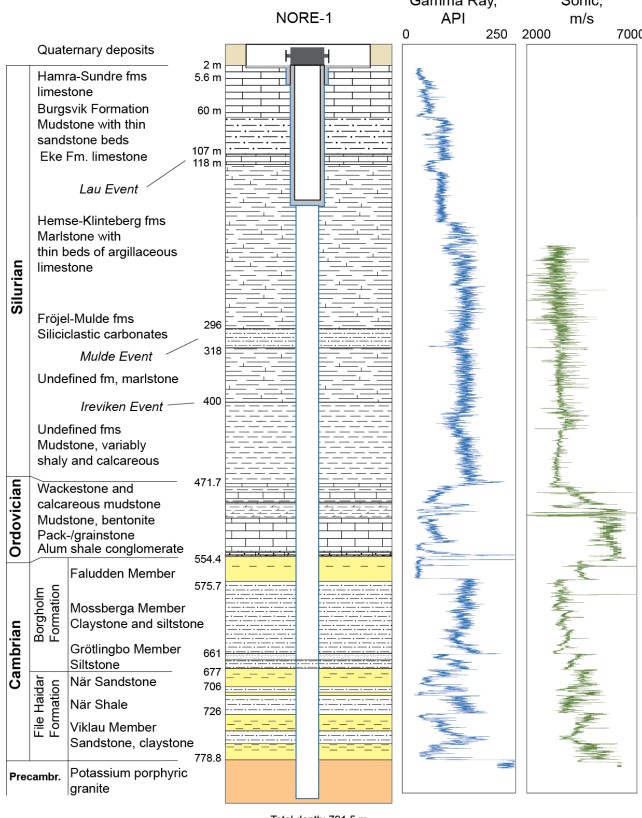

**Figure 2. Stratigraphy of the Nore-1 borehole. Note the limestone in the upper 60 m and the shaley sections down to about 500 m.**

In this paper we present some of the reflection seismic data acquired at Sudret along with results from geophysical logging and distributed acoustic sensing (DAS) from the two new cored boreholes drilled within the survey area (Nore-1 and Nore-2). During the processing it was observed that depth conversion using the normal moveout (NMO)

velocities resulted in a poor match with the known lithology and logs from the cored boreholes. Analysis of the DAS and acoustic logging data show that the NMO velocities are significantly higher than the DAS and logging velocities. We demonstrate with seismic modeling tests that this discrepancy can be explained by the upper c. 500 m of the bedrock sequence being anisotropic. Finally, we discuss the relevance of these studies for offshore $CO_2$ storage in the Baltic Sea area.



## 2   Geological setting and results from coring

The Cambro–Silurian succession on south Gotland comprises c. 225 m of Cambrian siliciclastics, c. 85 m of Ordovician carbonates and c. 470 m of Silurian marlstone-dominated strata (Fig. 2). Prior to drilling of the Nore-1 and Nore-2 wells, only a few rotary wells with scattered cored intervals have penetrated the entire Cambro-Silurian sequence on south Gotland.

The cored succession from 177 m and down to the basement at c. 790 m illustrates the characteristic Lower Paleozoic depositional evolution of the intracratonic Baltic Basin. The basin formed during the Late Ediacaran–Early Cambrian due to weakening of the central parts of the craton to the east when Baltica separated from the Rodinia supercontinent (Šliaupa et al. 2006). During most of the Cambrian and Ordovician relatively calm passive margin settings existed in the Gotland area. However, in the Silurian preceding the Caledonian collision between Avalonia and Baltica a significant phase of foreland subsidence occurred, which generated a substantial increase in the thickness of the Silurian.

The Cambrian siliciclastic sequence illustrates the progressive deposition linked to the overall transgressive nature of the Cambrian Sea. Stacked fining-upwards sequences grading from sand to clay are frequent in the sequence and interpreted to be formed in a shore-face and inner shelf dominated setting. The Lower Cambrian (Series 2) in the Nore wells is represented by the c. 100 m thick File Haidar Formation, which in turn is divided into the Viklau Member, När Shale and När Sandstone. The c. 50 m thick Viklau Member consists largely of well cemented quartz arenites with low porosity and permeability. Similarly, the c. 20 m thick När Sandstone is predominantly well cemented, with only a few meter-thick intervals with higher porosity and permeability. Intermediate beds are composed of dense silty mudstone. The overlying Borgholm Formation (about 120 m thick) includes the Grötlingbo, Mossberga and Faludden members and is dominated by variably silty mudstone. However, within this formation the c. 20 m thick medium-grained, porous and permeable Faludden sandstone occurs, which is arbitrarily equivalent with the Faludden Member (Fig. 2). This sandstone is the main potential storage reservoir for $CO_2$ in the south Baltic Sea where it reaches thicknesses exceeding 50 m as well as reaching depths of below 800 m. The Grötlingbo Member and parts of the Mossberga Member are dated to the upper Mioling (upper part of the lower Cambrian) while the remaining part of the formation is dated to the Furong. Much of the stratigraphic division on the Cambrian is based on the work by Nielsen and Schovsbo (2007, 2015).

It is notable that strata representing the Furongian Alum Shale Formation are almost completely absent on Gotland. Locally, as in the Nore wells, there is a conglomerate with alum shale fragments and glauconitic sandstone, which is a few decimeters thick. The absence of layers belonging to the Furongian is explained by exposure and extensive erosion in the area related to eustatic sea level changes (Nielsen and Schovsbo, 2015).

The Ordovician sedimentation on Baltica took place in a shallow epeiric sea surrounded by low relief and tectonically stable landmasses supplying limited amounts of terrigenous sediments into the depositional areas. The deposition was characterized by low sedimentation rates as well as periods of non-deposition, resulting in a condensed carbonate succession. The sediments are dominated by argillaceous limestone and calcareous mudstone with varying amounts of skeletal carbonate fragments and grains. Variegated red-brown grey wackestone and calcareous mudstone are the most common rock types. The Ordovician sequence is commonly about 80 m thick in wells on south Gotland. Locally, the thickness reaches 100-125 m due to the occurrence of carbonate mud mounds. The Ordovician is roughly divided into three parts, a lower 30-40 m thick grainstone- and packstone-dominated unit followed by an up to 20 m thick mudstone- and claystone- unit with thick bentonite layers. This unit can, under favorable conditions, generate a clear reflection on





seismic profiles on Gotland. The upper part of the Ordovician consists of dense red, brown and grey wackestone and
packstone with thin irregular layers of mudstone. This unit also contains carbonate mud mounds that can reach several
tens of meters in height, hence, the unit can locally be as thick as 70 m (Sivhed et al., 2004; Erlström and Sopher, 2019;
Levendal et al., 2019).

The Silurian succession has a total thickness of just under 500 m on south Gotland and covers about 10 million years
(428–418 Ma), corresponding to the Wenlock and Ludlow epochs. The original sediments were deposited on a shallow
shelf that covered large parts of the central Baltic Sea basin during Silurian time. The Silurian on Gotland illustrates
progressively deeper marine depositional conditions in a southwesterly direction with increasing amounts of marlstone
and mudstone in comparison to the north and northeast where biohermal limestone dominates.

A 55 m thick dark grey and shaly mudstone constitutes the basal most part of the Silurian on south Gotland. This is
followed by a c. 300 m thick relatively homogeneous marlstone-dominated sequence with only a few thin intervals with
greater amounts of limestone. The upper c. 120 m of the Silurian is more varied and composed of a mixed limestone-
mudstone-sandstone sequence belonging to the Eke, Burgsvik, Sundre and Hamra formations.

Note that the Ireviken (c. 400 m depth), Mulde (c. 310 m) and Lau (c. 110 m) events are coupled to global oceanic
changes (e.g. Calner et al., 2004) and that the geophysical log responses (Fig. 2) suggest that they may be reflective.
Specifically, in connection to these 10-40 m thick events, higher sonic wave velocities and lower gamma-ray (GR)
readings are often observed, reflecting a change in the depositional setting.



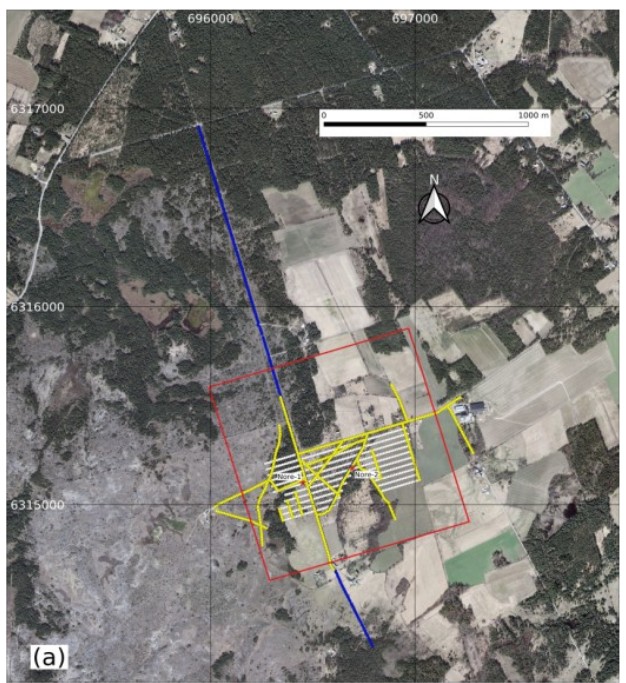

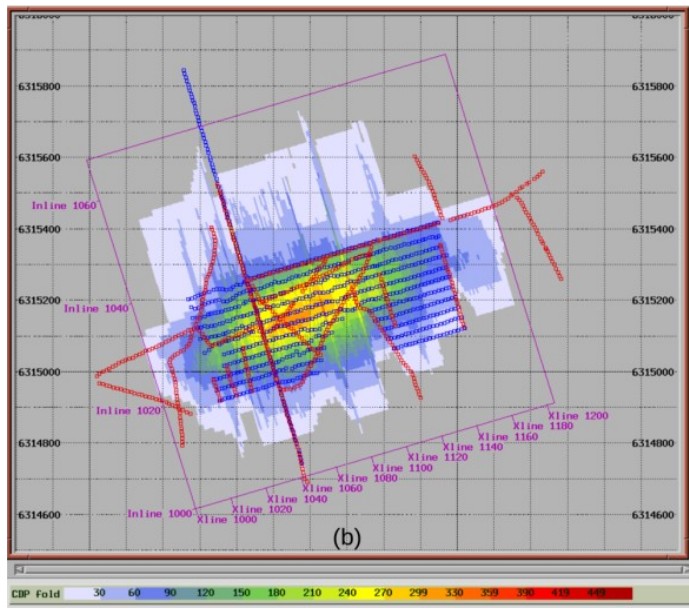

**Figure 3.** **(a) Source and receiver locations for the 3D acquisition, annotated with yellow and white circles, respectively. Source and receiver locations for the 2D acquisition are annotated with blue circles. The red rectangle outlines the 3D area. Red stars mark the Nore-1 and Nore-2 boreholes. (b) Detail of the 3D area with Inlines and Crosslines marked along with the fold for CDP bin sizes of 5 m in the Inline direction and 15 m in the Crossline direction. Blue and red circles show the locations of the receiver and source points, respectively. Aerial photo in (a) is from the Swedish Land Survey (lantmateriet.se). Coordinates are in the SWEREF99 TM system.**



### 3    Seismic data

#### 3.1    Acquisition

Both 3D and 2D data were acquired over a four day period in mid November, 2023. A skid steer loader mounted weight drop hammer (500 kg falling mass) with a base plate was used as a source and 410 nodal units were available for recording. In total, 704 receiver locations were occupied for the 3D with acquisition along 19 source lines (Fig. 3a). The 3D survey was done in two patches by moving  294 units to the new receiver lines (point locations) and reshooting the same source lines. Nearly all planned source locations were accessible with a total of 612 source points being accessed.

While the second round of source points was being shot the 116 nodal units that were not deployed in the 3D were moved to the central part of the 2D profile (Fig. 3a) in order to increase the coverage over the 3D area. Once the 3D survey was completed 213 nodal units were moved to the 2D profile. To further increase the coverage of the 3D area, 49 of these units were placed along one of the source lines leading to the Nore-2 borehole. Once the 3D acquisition was completed the 2D profile was acquired with 278 source points being shot. For both the 2D and 3D the hammer was

dropped 5 times at each source point. Timing for the impacts were recorded on a microsecond accuracy GPS time event recorder.

**Table 1. Processing steps applied to the 3D data**

| Step | Process |
|------|---------|
| 1 | Download nodal data and extract SEGD data given impact times |
| 2 | Read SEGD data and stack hammer hits |
| 3 | Add geometry |
| 4 | Spherical divergence correction, Tpower=1.0, max time=500 ms |
| 5 | Deconvolution, 5 ms gap, 150 ms filter length |
| 6 | High pass filter, 40-70 Hz |
| 7 | Brute stack |
| 8 | Residual statics |
| 9 | Median filtering |
| 10 | Bandpass filter, 30-50-150-250 Hz |
| 11 | Apply statics to fixed datum, 20 m at 3500 m/s |
| 12 | Mute air blast and data below |
| 13 | AGC, 200 ms |
| 14 | Velocity analysis |
| 15 | Residual statics |
| 16 | NMO |
| 17 | Stack |
| 18 | Trace balance |
| 19 | FXY deconvolution |



### 3.2 Processing

Given the Inline receiver spacing of 10 m and the Crossline spacing of 30 m it was decided to use 5 m by 15 m CDP bins for the processing. This led to Inlines ranging from 1000 to 1068 and Crosslines ranging from 1000 to 1204 (Fig. 3b), resulting in generally high fold within the receiver line locations, but low fold outside this area. Therefore, the images presented in this paper are most reliable within the 300 m by 700 m area covered by the receiver lines, specifically between Inline 1018 to 1038 and Crossline 1030 to 1170.

Source generated noise consists of coherent arrivals of direct or diving P- and S-waves, the air blast, as well as surface waves (Fig. 4a). In addition, 50 Hz noise is present at some geophone locations. A highpass 40-70 Hz filter removes much of the surface wave noise and attenuates the 50 Hz signals, leaving mainly linear coherent noise interfering with the reflections (Fig. 4b). At near offsets there is a rather weak first arrival with an apparent velocity of about 4500 m/s. This arrival is observed at somewhat varying velocity throughout the survey area and is interpreted as a wave propagating through a thin near surface high velocity layer. A later strong arrival at an apparent velocity of about 3500 m/s is clear (DW in Fig. 4a), as well as another one at about 1800 m/s (S in Fig. 4a). These two arrivals mask much of the reflected energy in the upper 500 ms. The faster one can possibly be explained as a diving wave propagating through the lower velocity rock below the 4500 m/s near surface layer. The 1800 m/s one may be a direct shear wave or represent a wave propagating through the water saturated uppermost loose sediments. In order to reduce the influence of these linear noise trains median filtering was applied to the source gathers. Prior to the filtering, an initial estimate of the residual statics was made and applied to increase coherency using the reflection window 300-500 ms. After median filtering a strong reflection package with a zero-offset traveltime of about 340 ms is clearly visible (Fig. 4c). Additional processing consisted of a tail mute following the air blast arrival, as well as AGC scaling prior to NMO and stacking. Iterative velocity analysis and residual statics were also performed. Greatest coherency in the reflections was achieved with stacking velocities on the order of 3400-3700 m/s from the near surface to 500 ms. The entire processing flow is given in Table 1.

Inline 1028 corresponds to the Inline that passes closest to the two cored boreholes (Fig. 3b), with Nore-1 located at about Crossline 1061 and Nore-2 located at about Crossline 1110. There is little lateral variation in the traveltimes to the reflections along Inline 1028 (Fig. 5a). Amplitude variations are significant, but some of this variation may be due to differences in fold and in amplitude versus offset effects, as well as variable interference from residual source generated noise that has not been completely removed. Reflections below about 450 ms are possible multiples that have not stacked out, but could also correspond to top basement reflections. A strong multiple at about 700 ms was attenuated by stacking at higher velocities. A Crossline passing over the boreholes (Fig. 5b) also shows only minor differences in traveltimes to the reflections, suggesting there is little structural variation below the 3D area.

The stacked volume was initially depth converted using a velocity of 3500 m/s (Fig. 6). This velocity corresponds to the average NMO velocity used. Depth conversion with the actual stacking velocities gives a similar result, with the exception that some of the later reflections appear deeper than when using the constant velocity. It is clear that there is a poor match between the geological section (Fig. 2) and the depth section (Fig. 6). The strong reflection at 340 ms is located at 600 m on the depth section and the reflectivity extends about 100 m below the top of the Precambrian, implying strong reflections in the upper part of the Precambrian basement when depth converting with the average NMO velocity.

Figure 4. Source gather processing. (a) Raw data with only trace balancing applied. (b) Same source gather as (a) with spherical divergence, deconvolution and a highpass 40-70 Hz filter applied. (c) source gather with processing up to step 10 in Table 1 applied. P: First arrival; DW: Diving P-wave; R: Reflection; SW: Surface wave; AB: Air blast; S: Direct shear wave.



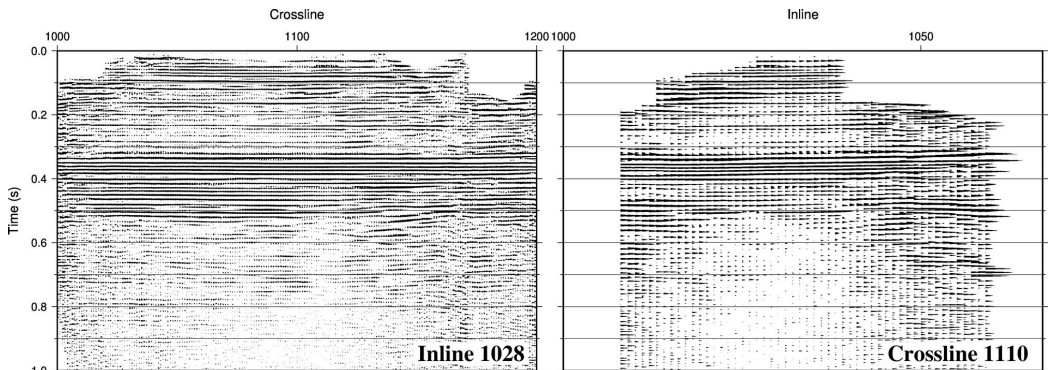

**Figure 5. Inline 1028 and Crossline 1110 time sections extracted from the 3D stacked volume.**

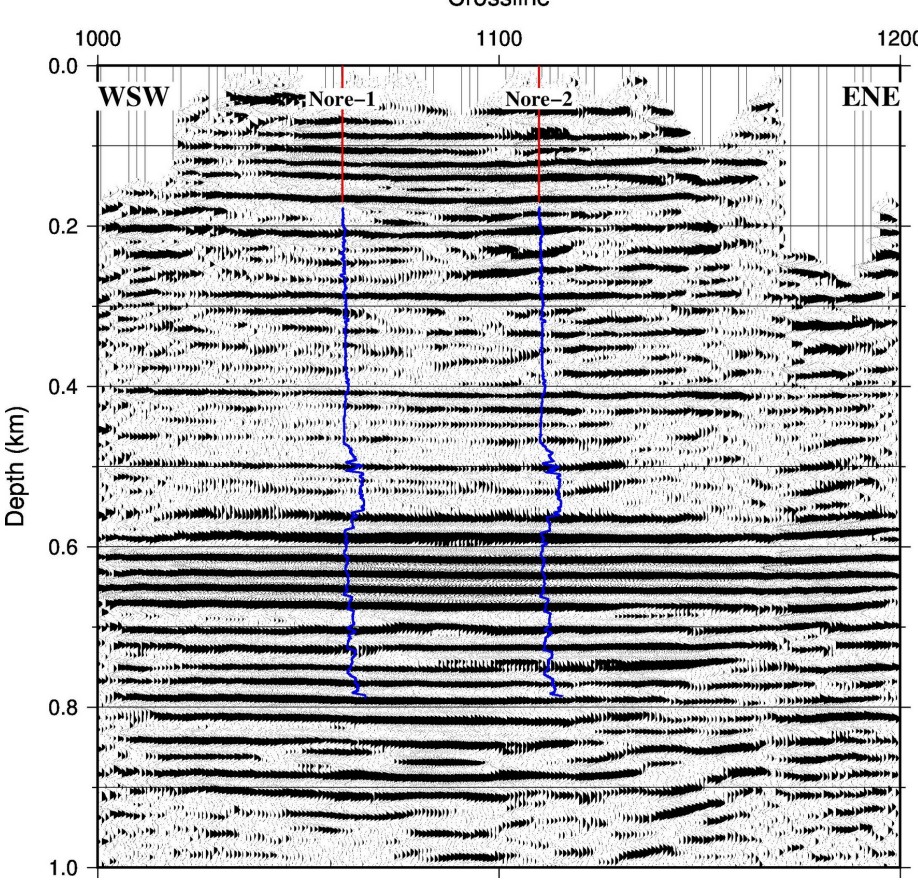

**Figure 6. Inline 1028 depth converted using 3500 m/s compared with the sonic logs from Nore-1 and Nore-2 (shown as blue and red lines). A 5 point boxcar filter was applied to the raw sonic logs before plotting.**




## 4    Borehole geophysical data

### 4.1    Distributed acoustic sensing data

Distributed acoustic sensing (DAS) was performed both in the Nore-1 and Nore-2 boreholes using a fiber optic cable
hanging loosely in the open hole. In Nore-1 data were recorded over the depth interval 17 m above to 475 m below sea

195   level. In Nore-2 the depth interval was 17 m above sea level to 720 m below. The fiber optic cable was sampled at 2.45
m intervals and data were recorded at a sampling frequency of 4000 Hz. A clear P-wave arrival is observed in Nore-1 on
source gathers close to the wellhead, as well as significant tube wave energy (Fig. 7a). Data from Nore-2 are dominated
by tube waves. In the Nore-2 borehole there are no clear P-wave first arrivals on near "zero-offset" raw source gather,
even after filtering tests at various frequencies. The tube waves dominate. However, when the source is offset some 20

200   to 100 meters then P-wave first arrivals can be discerned down to about 400 m below sea level (Fig. 7b). Even though
P-wave first arrivals cannot be identified deeper down their arrival times can be determined at the locations where tube
waves are generated. Given these tube wave generation depths it is found that the average velocity is no greater than
3100 m/s down to about 580 m below the surface (Fig. 7). Above this depth the average velocity is less.

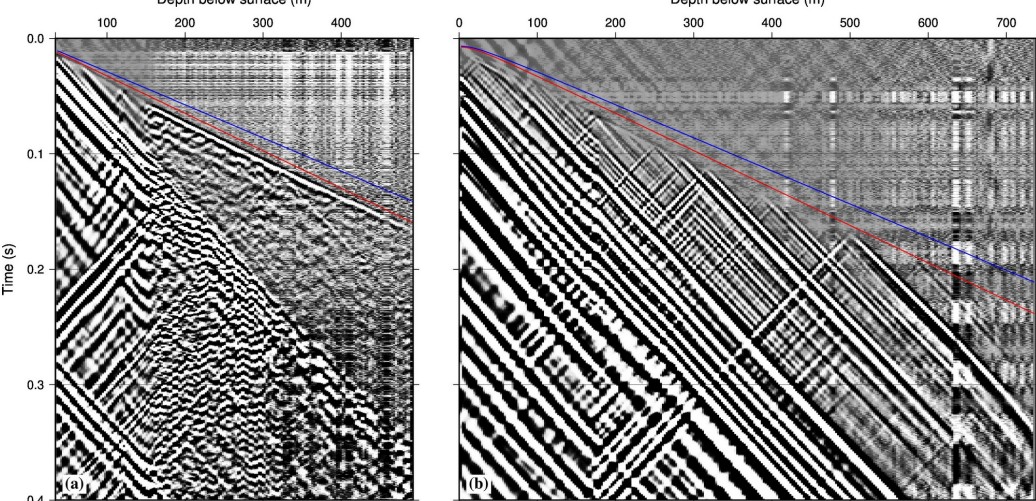

**Figure 7. (a) Near offset source gather recorded in the Nore-1 borehole. (b) Offset DAS source gather recorded in
the Nore-2 borehole with the source about 25 m from the wellhead. The red curve marks the expected first
arrival time for a wave propagating through a 3100 m/s constant velocity medium, while the blue curve gives the
arrival time for a wave propagating at 3500 m/s.**

It is well known that for many anisotropic layered sedimentary rocks that the horizontal velocity is generally greater
than the vertical velocity (e.g. Thomsen, 1986). Furthermore, in anisotropic media the NMO will also depend upon the
horizontal velocity, not only the vertical velocity. In the case of elliptical anisotropy the NMO velocity will correspond
to the horizontal velocity. Therefore, with the support of the DAS measurements a depth conversion velocity of 3100
m/s can be justified as a better estimate of the vertical velocity than the average NMO velocity of 3500 m/s. In the

following section we show that an average velocity of 3100 m/s is also consistent with sonic velocity measurements in
the boreholes.



### 4.2    Sonic log

After completion of drilling, geophysical logging data were acquired in the Nore-1 and Nore-2 boreholes. We focus here on the sonic logs (Fig. 8a). Due to casing installations in the boreholes, data are only available below about 170 m

depth. The sonic velocities generally fall between the average NMO velocity of 3500 m/s and the interval velocities from the DAS measurements. It is well known that high frequency sonic velocities are generally faster than those at seismic frequencies (Stewart et al., 1984). This can explain the discrepancy between the sonic and DAS velocity but cannot explain the difference between the sonic and NMO velocity. In order to compare the sonic log measurements with the surface seismic data a synthetic seismogram was generated based on the sonic data from the two boreholes.

Since no sonic velocity information was available for the upper 170 m we used the DAS data as a guide for this interval. Furthermore, a high velocity layer at the surface was included to account for the hard limestone found in the upper part of the borehole (Fig. 2). With the assumed velocity functions a good match is obtained between the observed data and the synthetic seismograms on Inline 1028 from the 3D survey (Fig. 8b). A more detailed comparison between the borehole data and the seismic data will be presented in the next section.

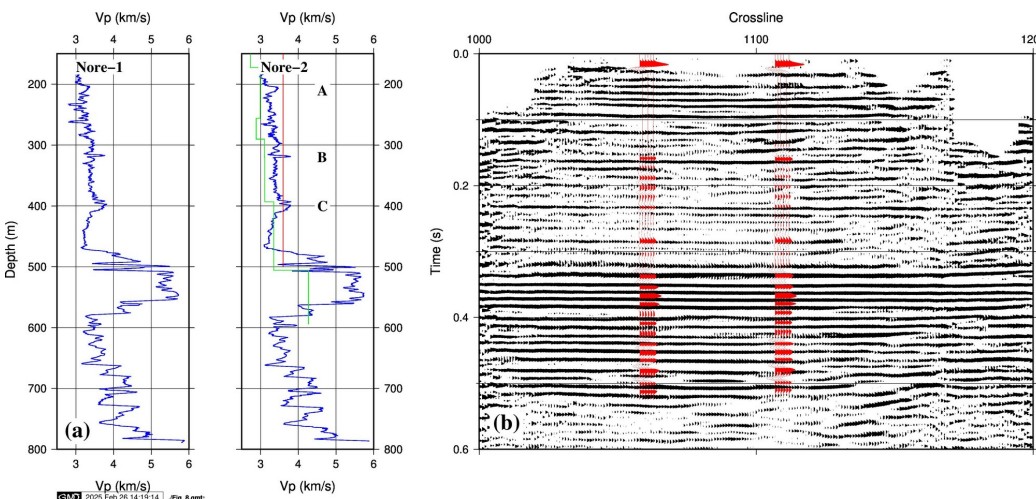


**Figure 8. (a) Sonic logs from the two boreholes. Green line in Nore-2 shows the velocities calculated from the DAS measurements, the red line shows the average NMO velocity to 500 m. A 5 point boxcar filter was applied to the raw sonic logs before plotting. (b) Synthetic seismograms from the two boreholes are plotted on Inline 1028 as a series of red waveforms.**

## 5    Discussion

### 5.1    Depth conversion

In order to compare the seismic data with the core it is more suitable to use a vertical velocity function based on the DAS data, rather than the NMO velocities. The deepest point on the DAS recording that provided a P-wave arrival time

was at about 580 m (Fig. 7b). Below this depth an average vertical velocity of 3100 m/s was used. It is now clear that the seismic data much better match the sonic logs (Fig. 9b). In fact, a number of reflections on Inline 1028 can be directly correlated to the borehole data. In particular the reflections at c. 210 m, 320 m and 400 m (coded A, B and C in Fig. 9b) are most likely generated by the relatively thin higher sonic velocity intervals at corresponding depths (coded A, B and C in Fig. 8). Reflections B and C probably correspond to the Mulde and Ireviken events, respectively, that



have been identified in the core. The gradual increase in sonic velocity starting at 470 m (top of the Ordovician) that continues for 20 to 30 meters is probably generating the low frequency reflection that marks the start of the high amplitude reflections below. Given that the high velocity limestone appears to have a rather constant velocity it is somewhat unexpected that several cycles of high amplitude reflections are present. However, there are variations in velocity within this interval that may be enough to generate the cyclic response. Support for this is seen in the synthetic

seismograms (Fig. 8b) which also show a cyclic response that agrees reasonably well with the observed data. Numerous reflections are present in the Cambrian sequence that contains the potential reservoirs. However, it is difficult to map a one to one correspondence between the sonic log or the core logging to the seismic depth section. This may be due to (i) impedance contrasts being low between these sandstones and the surrounding shales, (ii) interference or tuning between the reflections from the interfaces in the alternating sequence of sandstones and shales (iii) higher lateral variability in

the lithology within the Cambrian than in the overlying Ordovician and Silurian rocks, (iv) interference from internal multiples from the overlying Ordovician limestone or (v) a combination of these factors. Inspection of the seismic data do indicate lateral variability in the seismic response in this interval, suggesting that there is some lateral variations even if the two sonic logs are quite similar through the Cambrian. Note that there is no clear reflection off the top of the basement as might be expected based on the sonic logs.

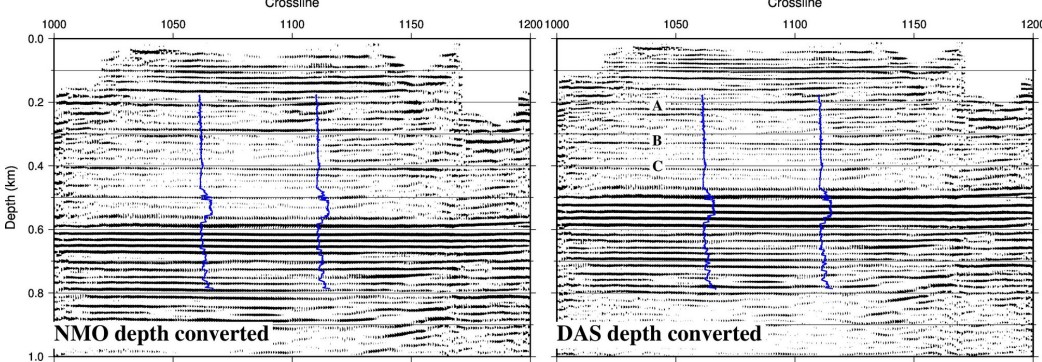

**Figure 9. Inline 1028 depth section depth converted with the NMO velocity (left) and the DAS velocity (right). The DAS depth converted section provides a better match with the sonic logs. A 5 point boxcar filter was applied to the raw sonic logs before plotting.**

### 5.2    Seismic modeling

In order to better understand the seismic response at the Sudret site a number seismic modeling tests were performed. As expected, we find that it is necessary to include anisotropy into the modeling to get a reasonable match between the real data and the modeled data. We show here results from two models. In the first model we use a velocity model based

on the sonic log. Due to the absence of sonic log data in the upper part of the boreholes we assume a 20 m thick high velocity layer at the surface and a DAS based velocity down to 170 m, while the velocity is constant in the crystalline basement. This first model is isotropic with the horizontal velocity equal to the vertical velocity, that is only the vertical velocity is considered in Figure 10. The second model is equivalent to the first, except that we assume the horizontal velocity is 13% faster than the vertical velocity in the upper 500 m, that is both the vertical and horizontal velocities are

considered in Figure 10. The anisotropy is also assumed to be elliptical and the $V_p/V_s$ ratio is assumed to be 2. Below 500 m we assume the rock is isotropic. Modeling was performed using the suea2df code, a 2D elastic anisotropic code available in Seismic Unix (Juhlin, 1995). Modeling results at offsets up to 1500 m are shown in Figure 11 along with a



filtered version of a typical source gather from the 2D profile. Three prominent arrivals marked by arrows show that the anisotropic modeling agrees much better with the real data than the isotropic modeling. What we interpret as a diving

wave (green arrow) propagating through the shaly/marly units above the Ordovician limestone arrives significantly earlier in the anisotropic modeling. The reflected P-P (red arrow) and P-S (yellow arrow) waves have the same near offset arrival times as in the isotropic and anisotropic modeling, but have moveouts with offset that much better match the anisotropic modeling. We conclude that the anisotropic model is consistent with the data, but other forms of anisotropy than elliptical cannot be ruled out.

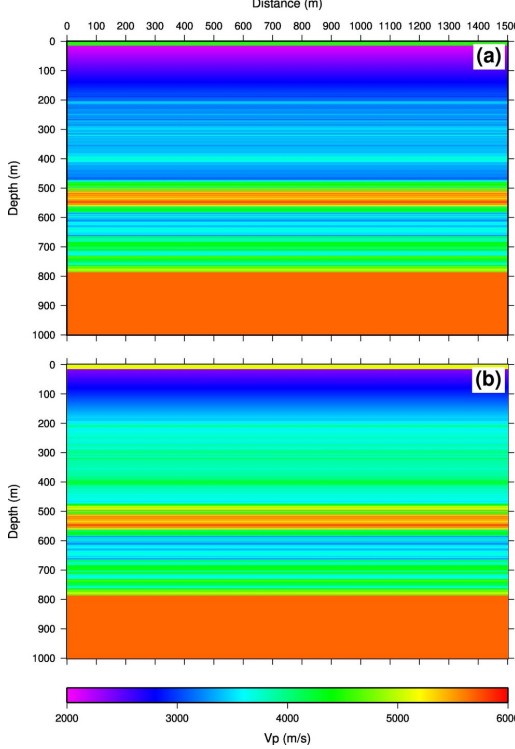


**Figure 10. Vertical (a) and horizontal (b) velocity models. For the isotropic model modeling only the vertical velocity was used for producing the source gather shown in Figure 11a. For Figure 11b the vertical and horizontal velocities were used assuming the upper 500 m is elliptically anisotropic.**

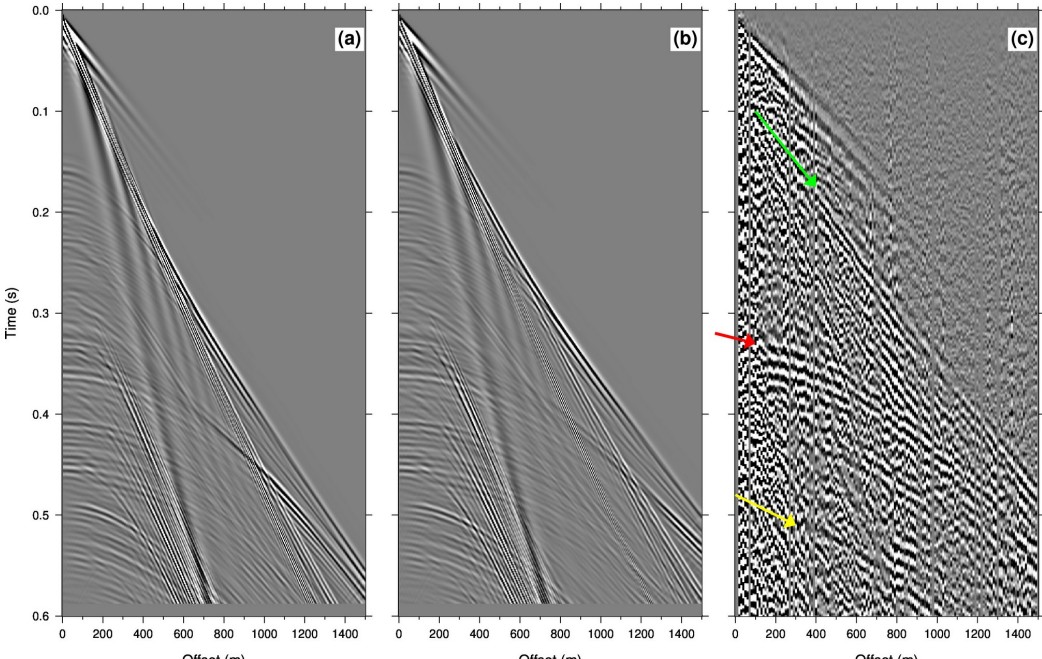

**Figure 11. (a) Results from isotropic modeling. (b) Results from anisotropic modeling. (c) Real source gather from the 2D seismic profile. Red arrow marks the reflection off the top of the high velocity Ordovician unit, yellow arrow marks the converted wave (P-S) reflection off the top of the Ordovician unit, green arrow marks the diving wave propagating through the rocks above the Ordovician unit.**

### 5.3 Relevance to offshore CO$_2$ storage

As shown in previous work (e.g. Sopher et al., 2016) the Ordovician limestone is a clear seismic marker that can be mapped over a wide area, both onshore and offshore. Below this unit we expect to find the Cambrian sandstones which

are the most prospective for potential CO$_2$ storage within Swedish waters below the Baltic Sea. Our study shows that it is difficult to identify reflections directly from these units even with the high frequency data that we have obtained. It is therefore unlikely that the sandstone layers can easily be mapped directly with seismic methods offshore. However, given the presence of the strong reflection from the Ordovician limestone and information from previous offshore drilling (Sopher et al., 2016) we can obtain a good estimate of the depth to the sandstone units. This is provided that we

know what velocity to use for depth conversion. It is likely that the offshore shaly/marly Silurian units are also anisotropic and if seismic data are used where there is no well control then this anisotropy should be considered prior to depth conversion. It can be worthwhile to revisit some of the more recent data acquired offshore in the Baltic Sea to investigate for the presence of anisotropy and to allow better depth estimates to be made in those areas.

### 6 Conclusions

Reflection seismic data were acquired on southern Gotland in order to gain a better understanding of the geological structure in the area and to extrapolate results from core drilling away from the boreholes as part of a program to investigate the potential for geological storage of CO$_2$ in the Baltic Sea area. During processing it became clear that there were significant discrepancies between the depth converted seismic data and the borehole data when the normal



moveout (NMO) velocity was used. The use of distributed acoustic sensing (DAS) data, along with sonic log measurements, led to a better understanding of the subsurface velocities, which improved the depth conversion and alignment with the core and sonic logs. Specifically, a vertical velocity function based on DAS measurements was shown to provide an image that better correlated with the borehole core and logs.

The discrepancy between the NMO and DAS velocities can be explained by the presence of anisotropy in the upper 500 m, with horizontal velocities being higher than vertical velocities. Seismic modeling showed that an elliptical anisotropic model with 13% anisotropy and a $V_p/V_s$ ratio of 2 could explain the observations. However, there may be other anisotropic models that can also be consistent with the data. Several marker horizons are observed in the seismic data that can be correlated to the boreholes when the data are properly depth converted. These include the strong reflection package from the Ordovician limestones, and the weaker reflections from the Mulde and Ireviken events. The target Cambrian sandstones show numerous reflections, but it is not possible to correlate these on a one to one basis with the individual sandstone units. This may be due to low impedance contrasts with the surrounding rocks, a complex interference pattern from thin sandstone layers and possible interference from internal multiples. Future processing of seismic data from the Baltic Sea area should take into account the presence of anisotropy and the complex reflection response from the Cambrian sandstones.

## Author contributions

PH, DS and CJ conceptualized and designed the data acquisition. BB and CJ were led the data acquisition and CJ was responsible for the data processing. ME and DS provided borehole data. CJ and ME led the geological interpretation. CJ and ME wrote the initial draft and all authors reviewed and contributed to it. All authors participated to the results discussion and approved the submission of this paper.

## Competing interests

The authors acknowledge that there are no conflicts of interest.

## Acknowledgments

This research was funded by the Geological Survey of Sweden.



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

## Figure captions


Figure 1. Location map. The red arrow shows the location of the survey area on southern Gotland.

Figure 2. Stratigraphy of the Nore-1 borehole. Note the limestone in the upper 60 m and the shaley sections down to about 500 m.

Figure 3. (a) Source and receiver locations for the 3D acquisition, annotated with yellow and white circles, respectively. Source and receiver locations for the 2D acquisition are annotated with blue circles. The red rectangle outlines the 3D area. Red stars mark the Nore-1 and Nore-2 boreholes. (b) Detail of the 3D area with Inlines and Crosslines marked along with the fold for CDP bin sizes of 5 m in the Inline direction and 15 m in the Crossline direction. Blue and red circles show the locations of the receiver and source points, respectively. Aerial photo in (a) is from the Swedish Land Survey (lantmateriet.se). Coordinates are in the SWEREF99 TM system.


Figure 4. Source gather processing. (a) Raw data with only trace balancing applied. (b) Same source gather as (a) with spherical divergence, deconvolution and a highpass 40-70 Hz filter applied. (c) source gather with processing up to step 10 in Table 1 applied. P: First arrival; DW: Diving P-wave; R: Reflection; SW: Surface wave; AB: Air blast; S: Direct shear wave.


Figure 5. Inline 1028 and Crossline 1110 time sections extracted from the 3D stacked volume.

Figure 6. Inline 1028 depth converted using 3500 m/s compared with the sonic logs from Nore-1 and Nore-2 (shown as blue and red lines). A 5 point boxcar filter was applied to the raw sonic logs before plotting.


Figure 7. (a) Near offset source gather recorded in the Nore-1 borehole. (b) Offset DAS source gather recorded in the Nore-2 borehole with the source about 25 m from the wellhead. The red curve marks the expected first arrival time for a wave propagating through a 3100 m/s constant velocity medium, while the blue curve gives the arrival time for a wave propagating at 3500 m/s.


Figure 8. (a) Sonic logs from the two boreholes. Green line in Nore-2 shows the velocities calculated from the DAS measurements, the red line shows the average NMO velocity to 500 m. A 5 point boxcar filter was applied to the raw sonic logs before plotting. (b) Synthetic seismograms from the two boreholes are plotted on Inline 1028 as a series of red waveforms.

Figure 9. Inline 1028 depth section depth converted with the NMO velocity (left) and the DAS velocity (right). The DAS depth converted section provides a better match with the sonic logs. A 5 point boxcar filter was applied to the raw sonic logs before plotting.


Figure 10. Vertical (a) and horizontal (b) velocity models. For the isotropic model modeling only the vertical velocity was used for producing the source gather shown in Figure 11a. For Figure 11b the vertical and horizontal velocities were used assuming the upper 500 m is elliptically anisotropic.


Figure 11. (a) Results from isotropic modeling. (b) Results from anisotropic modeling. (c) Real source gather from the 2D seismic profile. Red arrow marks the reflection off the top of the high velocity Ordovician unit, yellow arrow marks the converted wave (P-S) reflection off the top of the Ordovician unit, green arrow marks the diving wave propagating through the rocks above the Ordovician unit.