# Peer review of "Reflection seismic investigations on south Gotland, Sweden, to evaluate CO2 storage strategies"

_EGUsphere, 2025_

## Author Response (AR1)

**RC1: 'Comment on egusphere-2025-938', Niklas Kühne**

General Comments

The aim of the study is to explore potential geological structures for CO2 storage (CCS). Several active seismic measurement methods were combined for this purpose. In addition to a 2D/3D seismic survey, geophysical logging and DAS measurements were carried out on two cored boreholes. The study area is located in the south of Gotland, Sweden. The 2D/3D seismic data is used to extrapolate information from the borehole cores over a larger area. The borehole measurements serve to calibrate the depth of the seismic sections. After conventional data processing, the reflectors of the depth-converted seismic sections do not correlate with the sonic logs and the geological sections of the two boreholes located in the study area. The discrepancy is attributed to an elliptical anisotropy in the upper 500 m. This results in differences between the horizontal velocity derived from the NMO correction and the vertical velocity required for depth conversion. The DAS measurements in the borehole provide an estimate for the vertical velocity. The application of the vertical velocity results in a high agreement between the reflectors and the sonic logs. The hypothesis of prevailing anisotropy is supported by modelling synthetic data and comparison with measured source gathers. The paper illustrates a successful combination of different measurement methods (surface seismic, borehole logging). It also shows how DAS technology can add value to conventional seismic methods. It emphasizes the importance of complementary use of seismic measurement methods to local boreholes in order to gain a large-scale understanding of the geology of potential CCS reservoirs.

The geology in the south of Sweden has already been examined in other studies. The study shown here clearly demonstrates the previously underestimated effect of anisotropy, which influences seismic velocities. The consideration enables a more precise depth estimation of measured reflectors, which is of crucial importance for CO2 storage. This finding is highly relevant for the evaluation of future data sets in this region and should also lead to a critical examination of results from older measurement campaigns. The use of DAS in the borehole is presented as a suitable means of determining the vertical velocity of seismic waves. The measurement can be carried out in parallel to the ongoing surface seismic survey and requires only minimal data processing in the study presented. As a result, information that improves the accuracy of the surface seismic was obtained at a manageable additional cost. Due to the small lateral velocity differences of the horizontally layered sediments, the DAS measurements also provide a plausible estimate away from the borehole. It remains to be seen in subsequent studies to what extent the approach used can be applied to more complex geology.

**Response: We thank the reviewer for careful reading of the manuscript and the following comments which will be addressed in the revised version.**

Specific Comments

The following section contains some questions/issues that arose while reading the text. The majority of these are questions of understanding and suggestions to make the paper even clearer. I only see a need for optimization in places and recommend accepting the paper with minor revisions. I go through the individual sections of the text chronologically and indicate the line number in the manuscript for better comprehensibility.

1. Introduction

I suggest explaining the term 'supercritical' (Line 58) and including a suitable source to clarify the necessary pressure-temperature conditions and typical depth ranges.

**Response: We added the definition of supercritical CO2 "(temperatures above c. 31 °C and pressures above c. 7.4 MPa)**

Figure 2 shows the stratigraphy of the Nore-1 well very clearly. Why was Nore-2 not shown with the corresponding borehole logs? If there are only minor differences between the two images, then one image is sufficient. This fact should then be mentioned in the text. I would like to see an explanation in the caption of Figure 2 as to why logging results are not available at all depths. I also recognize the distinction between the cased borehole and the cored area in the illustration. A labelling would additionally support the distinction.

**Response: Figure 2 is revised and includes now labeling of the well design. A phrase explaining that the two wells are more or less copies of each other and hence only Nore-1 is used in the illustrations.**

2. Geological setting and results from coring

I am somewhat confused about the depth of the cores described in Line 80. Can you please explain if both holes were cored for the identical depths? I am also interested in whether the stratigraphy in Figure 2 was obtained directly from the cores and if so, where the information for the upper 170 m is derived from in this case.

**Response: An explanation to the coring and well design is now added. References are given to the the lithostratigraphical division, including the upper section.**

I find the description of the geology very interesting and comprehensible. However, my personal perception is also that the description is very detailed and provides information that is not necessary for the other sections of the paper. As an example, I would mention the section Line 87-101, where the Cambrian sequence is described in great detail and accuracy. I consider it sufficient for the whole Cambro-Silurian sucession to explain the history of their formation with a focus on the reflective layers and potential CO2 storage areas, as has already been done very clearly in large parts of the section.

**Response: The text has been altered and not as detailed as before to fit with what the reviewer suggests.**

In contrast to the other figures, the resolution of Figure 3 is lower and should be adjusted. For a better comparison between a) and b), I also recommend adapting the different colours for the source/receiver positions to each other (a): yellow/white, b) blue/red). In addition, the label 'b)' should be displayed in the same style and position as in image a). Finally, the position of the drill holes could also be included in Figure b).

**Response: Resolution of entire Figure 3 has been improved and Figure 3b has been redone according to the recommendations.**

3. Seismic data

3.1 Acquisition

I find the description of the measurement procedure and the repositioning of the receivers in several steps confusing (lines 135-146). I suggest a clearer description of the process and/or visualization in a map or diagram with changing number and position of receivers used.

**Response: We now show in Figure 3b the two deployments of the receivers and have reworded the text as follows: "The 3D survey was done in two patches by first deploying 410 units on the northernmost receiver lines (deployment #1 in Fig. 3b) and then moving 294 of these units to the remaining receiver lines and placing the other 116 units along the central parts of the 2D profile (deployment #2 in Fig. 3b ). This required reshooting the same source lines (yellow circles in Figure 3) to get complete coverage.**

3.2 Processing

I see the listing of the processing steps in Table 1 and the illustration of the influence on the data in Figure 4 as positive.

In lines 179-185, the discrepancy between the geological section and the section converted with the NMO velocity is discussed. The argument could be supported visually by integrating the geologic section into Figure 6 and highlighting the depth differences of the prominent reflectors with arrows. Could you please also refer to the red lines in Figure 6 and explain where these constant values for the upper borehole section come from.

**Response: We now write in the figure caption: "Inline 1028 depth converted using 3500 m/s compared with the sonic logs from Nore-1 and Nore-2 shown as blue curves. The red lines down to 170 m mark the cased sections of the boreholes that were not possible to log."**

**We do not find it useful to add the geological section into Figure 6. The figure clearly show that the strong reflectivity at 0.6 to 0.7 km is not connected to anything in the sonic logs and that there appears to be a depth discrepancy of about 100 m.**

I have a few more general questions about the extensive and comprehensible processing flow. If I understand the description of the processing correctly, it was done exclusively in the CSG domain. Was a repetition of some processing steps in CRG domain or Inline/Crossline domain considered? If not, what were the arguments against this? In addition, no migration of the data was applied. I would like to see a brief explanation of this decision in the text.

**Response: Data were processed in the CSG domain with some checking of results in the CRG domain. Processing in the CRG domain would have introduced artifacts given the highly irregular configuration of the source lines. FXY decon was applied on the Inlines and Crosslines. We chose not to present migrated sections since the area is essentially 1D. Furthermore, the irregular fold introduces artifacts into the migrated volume. We have added the following sentence to the text: "Given the 1D structure of the area and the variable fold we have chosen not to present migrated results since migration introduces some artifacts into the volume."**

4. Borehole geophysical data

4.1 Distributed acoustic sensing data

I wonder what influence the loosely hanging cable has on the coupling and therefore the data quality. Could you please provide a study that has dealt with this question? In the text, the data is

referred to 'sea level' (line 194/195/200), whereas in Figure 7 it is referred to 'depth below surface'. If the two reference depths differ in the study area, the values and terminology would have to be harmonized. Lines 196-198 address problems with the quality of the DAS data. Apart from frequency filtering, are there other strategies for avoiding tube waves during acquisition or processing?

**Response: Generally, for best DAS VSP results the fiber cable should be (1) cemented behind the well casing (permanent installation), (2) securing the cable by clamping it to the production tubing within the casing or (3) by deploying the cable loosely inside the borehole using wireline or slickline method (Munn et al. 2017). Different studies have demonstrated that tensing the cable improves the coupling, hence the signal quality, while others show the opposite. For example, Henninges et al. (2021) discuss the effect of tensing the cable to improve the coupling to the borehole wall versus slacking it by 1-20 m (releasing the tension by adding more cable to the borehole - loose suspension) and conclude that the tensed one produced data with the highest SNR. On the hand, Constantinou et al. (2016) report the opposite, where the SNR increased by adding additional slack making the fiber loosely suspended inside the well. Although this effect was evaluated numerically (see Schilke et al. (2016) or Henninges et al. (2021)), it is our experience that this effect is highly case specific and related to the well verticality (deviated well or not; in the case of the deviated wells, tensing the fiber along the azimuth following the well inclination) or other factors related to borehole conditions.**

**We added the sentence "Ideally, the fiber optic cable should be cemented in behind casing or in the borehole itself, but this was not an option for these wells since they were to be re-entered for further studies."**

**Munn et al. (2017): www.sciencedirect.com/science/article/abs/pii/S0926985117300228)**
**Constantinou et al. (2016): library.seg.org/doi/10.1190/segam2016-13950092.1**
**Schilke et al. (2016): library.seg.org/doi/10.1190/segam2016-13527500.1**
**Henninges et al. (2021): se.copernicus.org/articles/12/521/2021/**

**Filtering the data removes some of the tube waves, but does not improve the P-wave data significantly. It also introduces some artifacts into the P-wave data. Since we are mainly interested in the traveltimes of the P-waves we choose to present the raw DAS data.**

**All depths in Figure 7 now refer to depth below surface.**

The description of the average velocity with increasing depth should be more comprehensible (lines 202-203). So that it becomes clear in which range the average velocity is 3100 m/s and from where it starts to decrease.

**Response: We now refer to depths in the DAS data to below surface for clarity, rather than below sea level. We have also added a reference to Figure 8a which shows the DAS interval velocities.**

5. Discussion

5.1 Depth conversion

Could you please clarify exactly which velocities from the DAS data were used for the depth conversion (line 229-230)? It is unclear to me whether the velocity below 580 m is constant at 3100 m/s or whether the average of all interval velocities in this range is 3100 m/s.

**Response: The DAS interval velocities in Figure 8a were used for depth conversion down to 580 m. Below this depth the average velocity was assumed to be 3100 m/s. This is lower than we see on the sonic log, but we also note that the sonic velocities are higher than the DAS velocities down to 580 m. The choice of 3100 m/s for this interval results in a reasonable correspondence between the synthetic seismograms and the surface seismic data (Figure 8b). We now write: "Lacking deeper DAS data, we assumed an average vertical velocity of 3100 m/s below 580 m."**

5.2 Seismic modelling

You conclude in line 268-269 that other forms of anisotropy could explain the data. Would for the case of elliptical anisotropy other combinations of the strength of the anisotropy and the thickness of the anisotropic region also explain the data? And if so, why did you choose the variant presented in the paper?

**Response: Several of the more complex anisotropic models could also explain the data. However, elliptical anisotropy is the most simple model that explains the observations and we therefore use it in the modeling. We cannot rule out that the true anisotropy is more complex. We now write: "We did not test more complex forms of anisotropy that could also explain the reflection moveout since we cannot verify if they are present or not. Therefore, we have assumed elliptical anisotropy, which is the simplest form of anisotropy that can explain the data. The horizontal Vp/Vs ratio is assumed to be 2."**

6. Conclusions

I find the conclusion consistent with the results presented and provides a successful link back to the introduction and research question. I would also like you to address possible limitations of the approach used in the paper when discussing the DAS application (lines 291-292). For example, is the use of vertical velocities from borehole DAS still an option for areas with greater lateral variation?

**Response: We have added the following text: "In the case presented here the structure is essentially 1D and DAS measurements in a single borehole is sufficient. For areas which have more lateral variations it may be necessary for DAS measurements in several boreholes to constrain the vertical velocities."**

Technical Comments

Line 1: Inconsistencies in the date format (6th and 13th) should be corrected.

**Response: Done**

Line 137: Addition of 'survey' after 3D.

**Response: Done**

144-145: Replace the colloquial description 'hammer was dropped 5 times' with a more formal expression.

**Response: We now write "Vertical stacking for both the 2D and 3D was 5 at each source point."**

225 (Figure 8): Remove a (presumably) unwanted overlay between the image description and the first image.

**Response: Done**

268 (Figure 10): Display figures side by side, adjust y-axis labelling from 'm' to 'km' as in other figures.

**Response: Done**

268 (Figure 10, caption): Possibly omit 'modelling' in the second sentence.

**Response: Done**

**RC2: 'Comment on egusphere-2025-938', Anonymous Referee #2, 20 May 2025**

The manuscript presents a valuable seismic investigation on southern Gotland to assess the $CO_2$ storage potential of Cambrian sandstones, integrating borehole data, DAS measurements, and seismic reflection profiles. The study offers useful insights into subsurface characterization in a region of strategic interest.

However, the manuscript would benefit from improved structure and clarity. The Introduction could be rewritten for smoother flow and clearer framing. Transitions between sections throughout the manuscript are at times abrupt, which affects overall readability. Enhancing the logical flow would improve the manuscript.

**Response: We have tried to take this comment into account when re-reading the text and have made some modifications that we hope improves readability.**

Technically, while the study identifies significant anisotropy (~13%) and uses DAS and sonic logs to refine depth conversion, the seismic processing workflow appears overly simplistic for the complexity of the subsurface. In particular, the absence of anisotropic migration or advanced velocity modelling limits the accuracy of structural interpretation and reflector positioning. Clarifying these limitations and discussing their implications would add value, especially given the study's aim to extrapolate to the offshore $CO_2$ storage.

**Response: We do not understand this comment. The subsurface is essentially 1D with very little structure. The reflection moveout is hyperbolic and the reflections are horizontal after NMO (see CDP gather below with constant NMO of 3500 m/s applied). Advanced processing will do little to improve the image and may introduce artifacts. We have added the sentence "Given the 1D structure of the area and the variable fold we have chosen not to present migrated results since migration introduces some artifacts into the volume."**

[Figure]

The comments below include section/line/figure specific suggestions and questions:

Line 55: What is considered a shallow injection depth here? What is the minimum and optimal depth typically required to maintain $CO_2$ in a supercritical state, specifically in Cambrian sandstone reservoirs?

**Response: We have the temperature and pressure condition for which CO2 is in the supercritical state.**

Line159: What would be a rough estimate of the velocity of the "near surface high-velocity layer" mentioned here?

**Response: The velocity was already given in the previous sentence, it is about 4500 m/s, but varies somewhat throughout the area.**

Line 174: Was any AVO analysis conducted to explain the mentioned AVO effect? Or the authors claim that these amplitude changes are more likely only attributed to acquisition and processing factors? Or is it anisotropy-related?

**Response: We did not do any AVO analysis, but it would be interesting to do so in the future given the clear reflection from the Ordovician limestone.**

Section 3.1 How was the 2D line relevant in the study?

**Response: We present a source gather from the 2D profile in section 5. We use the 2D data for comparison with the seismic modeling since we have longer offsets in the 2D which makes it easier to see the differences in moveout with offset.**

Section 3.2 and further: Given the identified anisotropy (~13%), was a migration (depth/anisotropic) considered during processing? Including it—or discussing its absence—would help clarify potential impacts on reflector positioning, structural interpretation and mismatch mentioned on line 183. The processing of the seismic data is very simple and lacks the consideration of anisotropy.

**Response: The anisotropy that is present does not result in the moveout being non-hyberbolic. Furthermore, the NMO velocities do not vary significantly and there is little structure. Therefore, standard seismic processing assuming flat lying interfaces is highly suitable for this data set. If we did not have the borehole data we would not be able to deduce that the media is anisotropic. We see no need to make things more complicated than necessary.**

Table 1: The velocity analysis is mentioned here. Please clarify whether a detailed velocity model was constructed across the 3D volume, or if a laterally constant velocity was assumed for stacking and depth conversion?

**Response: The velocity functions are similar throughout the area. We added the sentence: "Similar velocity functions were determined throughout the 3D volume and along the 2D profile."**

Section 5.2: Have you explored other anisotropic models beyond elliptical (e.g. VTI or orthorhombic) to test if they yield better matches with the DAS and seismic data?

**Response: We do not have the data to consider whether a more complex anisotropy may be present. There does not appear to be any azimuthal dependency on the NMO so VTI or orthorhomic are not likely. Elliptical anisotropy is the simplest form of anisotropy that is consistent with our observations so have we chosen to work with this. We have added the sentences: "We did not test more complex forms of anisotropy that could also explain the reflection moveout since we cannot verify if they are present or not. Therefore, we have assumed elliptical anisotropy, which is the simplest form of anisotropy that can explain the data."**

Section 6: The results were meant to be extrapolated to investigate a potential Baltic Sea area storage. How would you account for potential differences in depth, stratigraphy, compaction, pore pressure, etc. between the onshore and the deeper offshore target?

**Response: Extensive offshore (albeit mostly from the 1970s) seismic data and a number of wells provide some constraints on the offshore structure (Sopher et al, 2016). The stratigraphy remains fairly constant, but with a generally thickening of the units to the south. In particular, the Faludden sandstone thickens to up to 50 m. We can speculate that the Faludden and overlying caprock will have similar properties as found in the Nore wells. We have added the following sentences "Based on Sopher et al. (2016) the offshore stratigraphy south of Gotland is generally similar to that found in the Nore wells, but with the Faludden sandstone becoming thicker southwards. Lack of core does not allow the offshore properties to be investigated in detail, but we speculate that the properties of the potential reservoir sandstones and overlying caprock will be similar to what is found in the Nore wells."**

Fig2: The log curves appear blurry and difficult to read. Consider improving the resolution.

**The figure is now in high resolution tif**

Fig 3: Consider zooming in on (a) portion of the figure. The well tags are difficult to read, and it is unclear where the receiver and shot points overlap. Also, consider adding a note on source and receiver spacing in the figure's caption as binning details for the 3D survey are mentioned but not the acquisition spacing. Please consider adding the borehole locations to part (b) of the figure as well, as they are referenced in Line 171.

**Response: We have redone Figure 3b so it should now be clearer where the receivers and source points are. We added to the caption the following sentence: "For the3D source and receiver spacing was 10 m with a receiver line spacing of 30 m."**

---

## Author Response (AR2)

Reviewer #1

General Comments:

Lines 143-146: These sentences describe the transition of the nodal units from 3D to 2D measurement, but it could be clearer why 213 units were moved to the 2D line when 49 units are not located on the 2D profile. I interpret the section to mean that 164, not 213 units, were moved to the 2D profile.

**Response: We now write "Once the 3D survey was completed 164 nodal units were moved and placed along the remaining parts of the 2.8 km long 2D profile (N-S blue line in Fig. 3a with the central part coincident with a 3D source line) To further increase the coverage of the 3D area, the another 49 units were placed along one of the source lines leading to the Nore-2 borehole."**

Lines 209-211: I am not entirely clear on the difference between the statements in these two sentences ("average velocity is no greater than", "average velocity is less"). The second sentence with "Above this depth" refers, in my understanding, to depths from 0-580 m and thus to the same range as in the preceding sentence.

**Response: We now write "Given these tube wave generation depths it is found that the average velocity is no greater than 3100 m/s down to about 580 m below the surface (Fig. 7), as also can be seen from the green line in Figure 8a for the calculated DAS interval velocities."**

Technical Comments:

Since I am not a native English speaker, my grammatical suggestions should be verified.

Text

Line 141: Remove additional space after "deployment #2 in Fig. 3b".
Line 144: Missing punctuation at the end of the sentence.
Line 256: Third person singular, "do" should be changed to "does".
Line 257: Singular instead of plural, "variations" should be changed to "variation".
Line 261: "of" is missing after "number".
Line 276: Singular instead of plural, "sources gather" should be changed to "source gather".
Line 283: Word order, "much better matches" should be changed to "matches much better".

**Response: All above changed.**

Figures

Figure 3, Caption Line 2: Missing space in "the3D".
Figure 4, Caption Line 2: Capitalize "Source" at the beginning of the sentence.

**Response: All above changed.**